# Quantitative Prediction of Human Pharmacokinetics and Pharmacodynamics of CKD519, a Potent Inhibitor of Cholesteryl Ester Transfer Protein (CETP)

**DOI:** 10.3390/pharmaceutics11070336

**Published:** 2019-07-15

**Authors:** Suein Choi, Seunghoon Han, Sangil Jeon, Dong-Seok Yim

**Affiliations:** 1PIPET (Pharmacometrics Institute for Practical Education and Training), College of Medicine, The Catholic University of Korea, Seoul 06591, Korea; 2Department of Pharmacology, College of Medicine, the Catholic University of Korea, Seoul 06591, Korea; 3Q-fitter, Inc., Seoul 06199, Korea

**Keywords:** CETP inhibition, dyslipidemia, cholesteryl ester transfer protein, pharmacodynamics, pharmacokinetics, in vivo-in vitro extrapolation (IVIVE), first-in-human, allometric scaling, prediction

## Abstract

CKD519, a selective inhibitor of cholesteryl ester transfer protein(CETP), is undergoing development as an oral agent for the treatment of primary hypercholesterolemia and mixed hyperlipidemia. The aim of this study was to predict the appropriate efficacious dose of CKD519 for humans in terms of the inhibition of CETP activity by developing a CKD519 pharmacokinetic/pharmacodynamic (PK/PD) model based on data from preclinical studies. CKD519 was intravenously and orally administered to hamsters, rats, and monkeys for PK assessment. Animal PK models of all dose levels in each species were developed using mixed effect modeling analysis for exploration, and an interspecies model where allometric scaling was applied was developed based on the integrated animal PK data to predict the human PK profile. PD parameters and profile were predicted using in vitro potency and same-in-class drug information. The two-compartment first-order elimination model with Weibull-type absorption and bioavailability following the sigmoid *E*_max_ model was selected as the final PK model. The PK/PD model was developed by linking the interspecies PK model with the *E*_max_ model of the same-in-class drug. The predicted PK/PD profile and parameters were used to simulate the human PK/PD profiles for different dose levels, and based on the simulation result, the appropriate efficacious dose was estimated as 25 mg in a 60 kg human. However, there were some discrepancies between the predicted and observed human PK/PD profiles compared to the phase I clinical data. The huge difference between the observed and predicted bioavailability suggests that there is a hurdle in predicting the absorption parameter only from animal PK data.

## 1. Introduction

Dyslipidemia is defined as an elevated plasma low-density lipoprotein (LDL) or triglyceride (TG) level or a low plasma high-density lipoprotein (HDL) level [1]. Dyslipidemia is a major risk factor for atherosclerotic cardiovascular disease (CVD) and increases the risk for an ischemic cerebrovascular accident. Large observational studies have reported a strong graded relationship between high LDL and/or low HDL levels and risk for atherosclerotic CVD [1,2,3]. Hydroxymethylglutaryl coenzyme A reductase (statin) is used as the gold standard for LDL-lowering therapy and has been shown to reduce the risk of CVD in humans. However, even after the aggressive use of LDL-lowering drugs, the incidence of residual cardiovascular events remains high, which suggests the need to find additional approaches [3].

Cholesteryl ester transfer protein (CETP) is a plasma glycoprotein that lowers HDL concentration and increases the concentrations of both LDL and very low-density lipoprotein (VLDL) by mediating and regulating the transfer of cholesterol ester from HDL and TG to the apoprotein B-containing lipoproteins, LDL, and VLDL [4]. Recent epidemiological evidence shows that increasing HDL cholesterol may reduce the risk of CVD independent of any change in LDL concentration [1,2,3,4,5]. Therefore, CETP has become a target for the treatment of dyslipidemia, and CETP inhibitors are expected to be a powerful class of drugs for increasing HDL and decreasing LDL levels, which should help reduce the risk of atherosclerotic CVD [4,6,7].

CKD519, a selective inhibitor of CETP, is undergoing development as an oral agent for the treatment of primary hypercholesterolemia and mixed hyperlipidemia. Preclinical in vitro and in vivo efficacy studies show that CKD519 is as potent as the comparator agent anacetrapib (half maximal inhibitory concentration (IC_50_) of CKD519 = 2.3 nM and IC_50_ of anacetrapib = 1.3 nM (in-house data)) and has no off-target effects. Although the development of investigational CETP inhibitors torcetrapib (Pfizer Inc., New York, NY, USA) and anacetrapib (Merck & Co., Inc., Whitehouse Station, NJ, USA) were discontinued due to their off-target effects and insufficient efficacy [8,9,10,11,12,13,14,15,16,17], several compounds including BAY 60-5521 (Bayer Inc., Leverkusen, Germany) are currently being investigated in preclinical and clinical studies [18,19,20,21]. Studies of these drugs have shown desired outcomes such as changes in LDL and HDL levels, and a possible lowering of CVD risk [11,21].

Translation of preclinical evidence to clinical development can be performed by quantitatively integrating in vitro and in vivo animal pharmacokinetics (PK)/pharmacodynamics (PD) information and extrapolating the parameters using allometric scaling and PK/PD modeling analysis [22,23]. Allometric scaling is a well-developed and widely accepted approach for the interspecies extrapolation of PK parameters [24,25,26,27]. The PK/PD modeling and simulation method is being increasingly applied at various stages of drug development and is recognized as a useful tool for reducing the uncertainty of predictions [28]. Additionally, the known PD profile of a comparator drug can be used to develop the human PK/PD model when there is a lack of solid PD studies [22].

This study aimed to use both methods in addition to the observed PD properties of a comparator drug (anacetrapib) to predict the appropriate efficacious dose of CKD519 for humans in terms of the inhibition of CETP activity.

## 2. Materials and Methods

### 2.1. Overall Strategy

Overall human PK/PD prediction was performed with the following procedures:Perform full PK samplings in three animal species to build-up animal PK datasets.Develop exploratory PK models by species from the datasets built in Step 1.Establish an overall (interspecies) PK model with allometric relationships for PK parameters in consideration of the model structure in Step 2.Human PK parameter prediction using the overall PK model.Human PK/PD modeling and simulation incorporating the PD information of the comparator drug.

### 2.2. Animal Full PK Study

Animal PK data for analysis were collected from preclinical studies performed at ChongKunDang laboratory. All animal experiments were performed with relevant guidelines and regulations, and the protocol was approved by the Institutional Animal Care and Use Committee (IACUC) of ChongKunDang (Approval number: S-13-015). A single dose of CKD519 was administered either orally or via intravenous infusion to hamsters (intravenous dose of 0.5 mg/kg, oral dose of 3, 15, or 45 mg/kg, n = 30); rats (intravenous dose of 0.5 mg/kg, oral dose of 5, 15, or 45 mg/kg, n = 20); and monkeys (intravenous dose of 0.1 mg/kg, oral dose of 1, 5, or 30 mg/kg, n = 16). The identical solution was administered to all animal species in the fasted state. Plasma CKD519 concentration was measured before and at 0.083, 0.25, 0.5, 1, 2, 4, 8, and 24 h after administration in hamsters and rats, and before and at 0.083, 0.25, 0.5, 1, 2, 4, 6, 8, 12, 24, 48, and 72 h after administration in monkeys. Two hamsters given identical doses were sampled in a rotating manner.

The plasma samples were stored at −70 °C before the analysis and were thawed at room temperature. As the sponsor did not agree to publish the detailed assay method, a brief description is given as follows. A sensitive, specific, and validated liquid chromatography coupled with a tandem mass spectrometer (HPLC-MS/MS) assay was used for the determination of CKD519 concentration in animal plasma samples. A structural analogue of CKD519 was used as the internal standard. The validated working range was from 2 ng/mL (the lower limit of quantification, LLOQ) to 20,000 ng/mL. Quality control (QC) samples were analyzed together with the study samples. Intra-day and inter-day precision and accuracy results were within the acceptance criteria based on the Ministry of Food and Drug Safety (MFDS) guidelines. An exploratory data analysis was performed through non-compartmental analysis (NCA) using the NonCompart package (developed by Bae, version 0.4.4) in R (version 3.5.1, The R Foundation, Vienna, Austria).

### 2.3. Animal PK Model Development

Using the PK data from each species, nonlinear mixed-effect modeling was conducted using NONMEM (version 7.4, Icon Development Solutions, Ellicott City, MD, USA) reflecting the clues from NCA. The first-order conditional estimation method with interaction (FOCE-I) was used whenever applicable. The adequacy of the model was checked using changes in the objective function value (OFV), visual inspection of various diagnostic plots (goodness-of-fit (GOF) plot), and methods of visual predictive check (VPC) and bootstrap were used for the diagnostics on the model stability and parameter reliability. The significance of model improvement was evaluated using a likelihood-ratio test (LRT). In the nested models, the result was considered statistically significant if the OFV decreased more than 3.84 (*p*-value < 0.05, degree of freedom (df) = 1) and 5.99 (*p*-value < 0.05, df = 2). The degree of freedom is defined as the gap of the numbers of parameters of the two subsequent models that are being compared. In the case of non-nested models, the Akaike information criteria (AIC) value was used to select the model that best described the data. R (version 3.5.1., R Foundation for Statistical Computing, Vienna, Austria) was used for data preparation, NCA, graphical analysis, model diagnostics, and statistical summaries.

Initially, a two-compartment model with first-order absorption was developed to describe the biphasic curve, and several absorption structures (e.g., zero-order with absorption lag time, Weibull-type absorption) were applied to the base model if needed. The change in absolute bioavailability (*F*) by dose levels was also considered. Each PK parameter was assumed to follow a log-normal distribution and is described as:*P*_i_ = *P*_TV_ × exp(η_i_),(1)
where *P*_i_ is the individual parameter for the *i*-th individual; *P*_TV_ is the typical value of the model parameter for the population; and η_i_ is the interindividual random effect following a normal distribution with a mean of 0 and variance of *w_i_*^2^ accounting for *i*-th individual’s deviation from the typical value *P*_TV_. Both an additive error model and a proportional error model were evaluated.

### 2.4. Incorporation of Allometry and Human PK Parameter Prediction

The volume parameters (*V***_c_**, *V***_p_**) estimated from the PK models for three different species were correlated through simple allometric scaling (=Body weight-dependent allometry) in the PK model fitted to the overall data from all species. This method is based on the power–law function, which can be represented as follows:Y = a × BW*^b^*,(2)
where Y represents the PK parameter of interest from each animal; BW represents the average body weight of each species from the relevant in vivo studies; and a and b are the allometric coefficient and exponent of the equation, respectively.

For the allometric scaling of the clearance parameters (*CL*, *Q*) in the same model, physiological parameters such as brain weight (BrW) or maximum lifespan potential (MLP) of each species were incorporated if necessary. The values for BrW and MLP used in this analysis are presented in Table 1 [29,30]. To address the uncertainty, two allometric relationships were selected as the final models, and two sets of human PK parameters were predicted: one from the allometry of best fit and the other from the most conventional simple allometry. The appropriateness of the extrapolated human PK parameter values was assessed by comparing them with those of the comparator. Since allometric scaling is not generally recommended for the absorption parameter, human absorption parameters were determined by the final estimates from the overall PK model instead of the allometric method.

### 2.5. Human PK/PD Modeling and Simulation

The plasma concentration (nM) versus time profile of CKD519 was simulated after single-dose administration of 5, 10, 25, 50, 120, and 250 mg for 72 h. Given that two sets of PK parameter values (one from the best fit and the other from the simple allometry) were suggested for humans, two sets of simulations were performed. In addition, PK/PD linking was performed using an exposure–response structure for the CETP inhibition obtained from the literature on anacetrapib as a comparator (% inhibition of CETP = (100 × C)/(*IC*_50_ + C), where C represents the plasma concentration of a CETP inhibitor and IC_50_ represents the plasma concentration where half of the maximum effect is achieved) [13,15]. The peak effect was around ~90% inhibition. Since no in vivo information for the *IC*_50_ value of CKD-519 was available, it was assumed using the known in vivo IC_50_ value for the comparator (22 nM) [15], and the observed difference in the in vitro potency between the comparator and CKD519 (IC_50_ 1.3 nM vs. 2.3 nM (in-house data), respectively). Anacetrapib was selected as the comparator for the in vitro potency study since detailed information including the in vivo IC_50_ value is known, unlike other CETP inhibitors. Considering this, CKD519 was assumed to be 1.7-fold less potent than the comparator (37 nM). Finally, the CETP inhibition (%) versus time profiles were simulated using two IC_50_ values (22 nM and 37 nM, reflected as 13.22 ng/mL and 22.24 ng/mL in the PD model, respectively), which corresponded to the PK profile simulated by dose levels. The efficacious dose was defined as the dose accomplishing the threshold in ≥50% of the population, which was determined using the median. Thus, the variability of the parameters was not incorporated in the simulation.

## 3. Results

### 3.1. Animal PK Dataset and Exploratory Data Analysis

The PK data from the animal species were obtained as planned. The observed plasma concentration–time profiles by species and dosage regimens are presented as Figure 1, and the animal PK properties of CKD519 are summarized as the results from NCA (Table 2).

These exploratory PK analyses showed that the plasma concentration–time profile of CKD-519 followed a biexponential disposition pattern, and its dose-normalized *AUC* decreased as the dose increased. In addition, we found clues for the necessity of an absorption model other than simple first-order absorption.

### 3.2. Animal PK Model by Species

The PK profile in each animal species was commonly best described by the two-compartment PK model with first-order elimination. Weibull-type absorption to describe the change in absorption rate with time and a sigmoid structure to explain the changes in *F* by dose were incorporated in all species. The PK model structure is graphically presented in Figure 2.

The differential equations for the structure were as follows:*F* = *F*_max_ × (1 − dose/(*F_50_* + dose)),(3)
*WB* = (*β*/*α*) × (t/*α*) *^β^*^− 1^,(4)
A_1,0_ = dose × *F*,(5)
dA_1_/dt = −*WB* × A_1_,(6)
A_2_/dt = *WB* × A_1_ − *CL/V_c_* × A_2_ − *Q/V_c_* × A_2_ + *Q/V_p_* × A_3_,(7)
A_3_/dt = *Q/V_c_* × A_2_ − *Q/V_p_* × A_3_.(8)
where *F* is the absolute bioavailability with *F*_max_ as the maximum bioavailability and *F_50_* as the dose reaching 50% of the maximum, respectively; *WB* is the time dependent absorption rate constant following Weibull distribution with *α* as the scale and *β* as the shape parameters of the Weibull distribution, respectively; A_1,0_ is the initial amount in the depot compartment; A_1_, A_2,_ and A_3_ are the drug amounts in the absorption, central, and peripheral compartments, respectively; *V*_c_ and *V*_p_ are the distribution volume of the central and peripheral compartments, respectively; and *CL* and *Q* are the elimination and intercompartment clearance, respectively.

All the PK profiles from each species were well-described using the model, and the final parameter estimates are presented in Table 3.

### 3.3. Overall (Interspecies) PK Model and Allometric Relationship

In simple allometric scaling, *V*_c_ and *V*_p_ values from the three species exhibited strong correlations (R^2^ = 0.9777, R^2^ = 0.9999, respectively), but *CL* and *Q* values had poor correlations (R^2^ = 0.5229, R^2^ = 0.8896, respectively). For clearance parameters, the allometry with the product of BrW showed the strongest correlations (R^2^ = 0.9760, R^2^ = 0.9993, respectively). Consistently, the relationship produced the best outcome in terms of OFV values and GOF when reflected in the overall PK model. The OFV values from models with various allometric relationships are presented in Table 4.

The final allometric relationship used in the overall PK model is as follows:Parameter value in each species = a × BW^b^ (for volume parameters, *V*_c_ and *V*_p_),(9)
Parameter value in each species = a × BW^b^/BrW (for clearance parameters, *CL* and *Q*),(10)
where BrW and BW represent the average brain weight and body weight of each species, respectively, and a and b are the coefficient and exponent of the allometric relationship, respectively.

As the exponents of the volume parameters initially estimated in the overall PK model were close to 1, and the parameters in three species showed strong correlations even when they were fixed to 1, we decided to fix them as 1 in the final overall PK model (no significant change in OFV was shown). The GOF plots of the final PK model are shown in Figure 3 The final parameter estimates from the overall PK model, and the precision of parameter estimates assessed by the bootstrap method are shown in Table 5.

### 3.4. Human PK/PD Simulation

As mentioned in Section 2.4, predicted human (with the body weight of 60 kg) PK parameter values were obtained using the simple and BrW-corrected (the best fit) allometric relationships and the final estimates for allometric coefficient and exponent in overall PK models. The sets of parameter values used for human PK/PD simulation are presented in Table 6.

The simulated plasma concentration–time profiles in a human after a single oral dose of 5, 10, 25, 50, 125, and 250 mg are shown in Figure 4. We could confirm that the area under the curve (*AUC*) and maximum plasma concentration (*C*_max_) did not increase proportionally with the dose in both cases because of the structural decrease in *F*. In the simulation results with BrW-adjusted parameter values, even with the same absorption properties, *T*_max_ was shorter, and *AUC* and *C*_max_ were lower than those with simple allometry parameter values. Plasma concentration values rapidly decreased within 12 h after dosing, and there was a clear distinction between the distribution phase and elimination phase when the parameter values from the BrW-adjusted model were used, probably because of the higher clearance (0.264 vs. 0.777) and shorter half-life along with higher intercompartment clearance (1.17 vs. 2.39). For doses of 5 mg and 10 mg, the concentrations remained higher than the IC_50_ for 7.5 h and 10.2 h in simulation from the BrW-adjusted model, and 10.5 h and 22 h in simulation from the simple allometry model, respectively. When the dose was higher than 25 mg, concentrations remained over the IC_50_ (= 37 nM) for 24 h, which is the predicted dosing interval in both simulations.

The simulated CETP inhibition (%)–time profiles for the previous PK simulation are given in Figure 5.

As shown in Figure 5, the pattern predicted for the inhibition of CETP paralleled the plasma concentration curve. Like the PK simulation profile in Figure 4, CETP inhibition (%) activity decreased faster in the BrW-incorporated model and showed distinctive changes within 12 h, whereas the simple allometry model did not show a distinctive change within 24 h. However, for doses of ≥25 mg, CETP activity was inhibited by >50% in every scenario and maintained for 24 h, which is the potential dose interval. Finally, for a dose of ≥125 mg, CETP activity was predicted to be inhibited by 100% (maximum effect), and the inhibition activity was maintained for at least the potential dose interval.

### 3.5. Prediction of the Human Efficacious Dose

From the simulation results, the daily administration of 25 mg appeared to reach the half maximal effective concentration and was maintained throughout the potential dose interval in every scenario. The PD response did not increase significantly with an elevated plasma concentration when the dose was ≥125 mg and reached the maximum inhibition effect (100%) for most of the day. Based on the literature on the comparator drug, maintaining CETP inhibition activity >50% for the dose interval was predicted to achieve a therapeutic efficacy for the HDL level [31,32]. Therefore, to satisfy the pharmacodynamic trough level for a clinical response with low plasma concentration, we concluded that the daily administration of 25 mg was the potential efficacious dose for humans.

## 4. Discussion

In this study, we predicted the human PK/PD parameters and profiles of CKD519 by integrating in vitro PD data and animal PK data with clinical PK/PD profiles of a comparator drug. From the torcetrapib study result, we assumed that the CETP inhibition should exceed the IC_50_ over a dosing interval to show clinical efficacy, which is the elevation of HDL [31,32]. Using an allometry scaling method, PK/PD modeling analysis, and the potency ratio of the in vitro IC_50_ to the known PD parameters of the comparator drug, we predicted the efficacious dose of CKD519 in humans and the limits of dose escalation in terms of the inhibition of CETP activity.

From the animal PK analysis, we selected the two-compartment first-order elimination model with Weibull-type absorption as the final model to describe the complicated absorption pattern. In the Weibull-type absorption model, the absorption rate constant is a time-dependent variable that follows the Weibull distribution, which is flexibly controlled by both the shape and scale parameter. Although it is more a descriptive model than a mechanistic model, in vivo drug absorption is a complex multistage process, and the flexibility inherent in the Weibull function may reflect the actual variable drug absorption rates changing along the gastrointestinal tract [33]. This model is often used as an alternative to describe the complicated absorption profile when simple-order kinetics cannot describe it adequately [34,35,36,37,38,39]. The first-order and zero-order framework failed to depict the absorption phases of CKD519 in the three species tested here, but these were well described by the Weibull-type absorption model, which showed that it was plausible to infer this model as a human absorption model.

Although the decrease in the dose-normalized *AUC* by dose can be explained by either an increase in *CL* or decrease in *F*, the terminal slopes in the log-scale graph for each species were similar at every dose, which implies that clearance remained unchanged as bioavailability decreased with the dose possibly due to saturation of the absorption pathway [40]. To describe the change in absolute availability by dose and to explain the absorption saturation, the sigmoid curve model with *F*_max_ and *F*_50_ was considered as the absolute bioavailability model, and the parameters of the model were estimated by integrating the PK data of all three species in this study. However, given the considerable physiological differences between species in the first-pass gut and liver metabolism, it is known that human bioavailability prediction based on animal PK data alone is inaccurate in many cases [41]. Additionally, a tablet formulation was used in the human study instead of a solution, which could have caused differences in bioavailability and absorption rate between animals and humans. As shown in Figure 6, discrepancies were found between the predicted and observed (first-in-human trial) human PK profiles, which were mostly due to the difference in bioavailability (detailed data cannot be shared as the first-in-human study has not been published yet). As bioavailability is mainly determined by intestinal absorption and first-pass metabolism, various physiologically-based in vitro–in vivo extrapolation (IVIVE) models reflecting the formulation difference have been developed to predict the human intestinal absorption rate and oral bioavailability [41]. Unfortunately, there were no other in vitro data or detailed formulation data that could be used to try these methods. The use of absorption IVIVE models rather than the simple integration of animal data would have shown better results in predicting the absorption parameters. This study showed the limitations of predicting absorption parameters only with animal PK data.

In this study, the exponent of clearance from the simple allometry was less than 0.55. According to the rule of exponent, simple allometric scaling tends to underestimate the parameters, and the use of correction factors is also controversial when the exponent of simple allometry is less than 0.55 [24,42]. However, as the clearance parameters (*CL*, *Q*) from the three animals did not show a good correlation when simple allometry was used (R^2^ = 0.5229, R^2^ = 0.8896, respectively) and because no reliable in vitro data was found such as microsome clearance, the use of correction factors (BrW, MLP) was considered on the allometric scaling model to improve the correlation and compensate for the underestimation. We tried to find which allometric scaling method explained the animal data best among the various allometric methods, and the scaling method using the product of BrW and the clearance parameters gave the strongest correlation between all species with a coefficient of determination (R^2^) in the range of 0.9780–0.9999. Predicted clearance values were higher when the correction factor was applied, which also supports the plausibility of using the correction factor on the overall model.

In this study, due to the lack of reliable in vivo PD data, we used the ratio of the in vitro IC_50_ of the comparator drug to that of CKD519 as the potency ratio to predict the in vivo IC_50_ value from the reference instead of using the observed in vivo IC_50_ of CKD519. As same-in-class drugs generally follow a similar pharmacodynamic pathway with different potency, PD information for a comparator drug can help predict the PD profile and develop the PK/PD model as well as extrapolate PD parameters from in vitro to in vivo studies. This lessens the uncertainty of the prediction and is more efficient in first-in-human trial designs, especially when the available data are insufficient or unreliable. A model’s predictability can be improved by observing biomarkers such as CETP activity or cholesterol level in a phase 1 study and applying the data in our model.

The absolute bioavailability was estimated for each dose (Table 7) by comparing the AUC of the observed oral PK profile in the first-in-human study and that of the predicted human i.v. PK profile from animal PK data. As shown in Table 7, the calculated bioavailability from the observed PK data decreased as the dose increased, implying the saturation of the absorption pathway. Even though the trends were similar, absolute bioavailability estimated by the observed PK profile was less than one-fifth of the predicted bioavailability using the animal PK data. Figure 7 shows the simulated PK/PD profile after applying the calculated bioavailability. The simulated PK/PD profiles with modified bioavailability were comparable to the observations in the first-in-human study, except for the slight under-prediction of plasma concentrations. Delayed absorption was also found, and the difference in formulation between animals and humans was considered to have caused the delay by affecting the absorption rate. Simulated *AUC*, *C*_max_, and *T*_max_ were within 2-folds, the range that is generally accepted for animal-to-human predictions. This result shows that the large discrepancy between the observed and predicted PK/PD profile could have resulted from the poor prediction of absolute bioavailability, and the PK/PD model developed only by animal data can be improved by using first-in-human study data.

## 5. Conclusions

This study predicted human PK/PD parameters and profiles by integrating in vitro and in vivo PK/PD data for CKD519 and clinical PK/PD profiles of the same-in-class drug. Allometric scaling, PK/PD modeling analysis, and in vitro IC_50_ were used for parameter prediction and model development. The prediction was unsuccessful, as a large discrepancy was highlighted from the observed profiles resulting from misprediction of the absolute bioavailability. This study also showed the limitations of human absorption parameter prediction using the animal data alone and the importance of incorporating the IVIVE method or first-in-human data in allometric scaling.

## Figures and Tables

**Figure 1 pharmaceutics-11-00336-f001:**
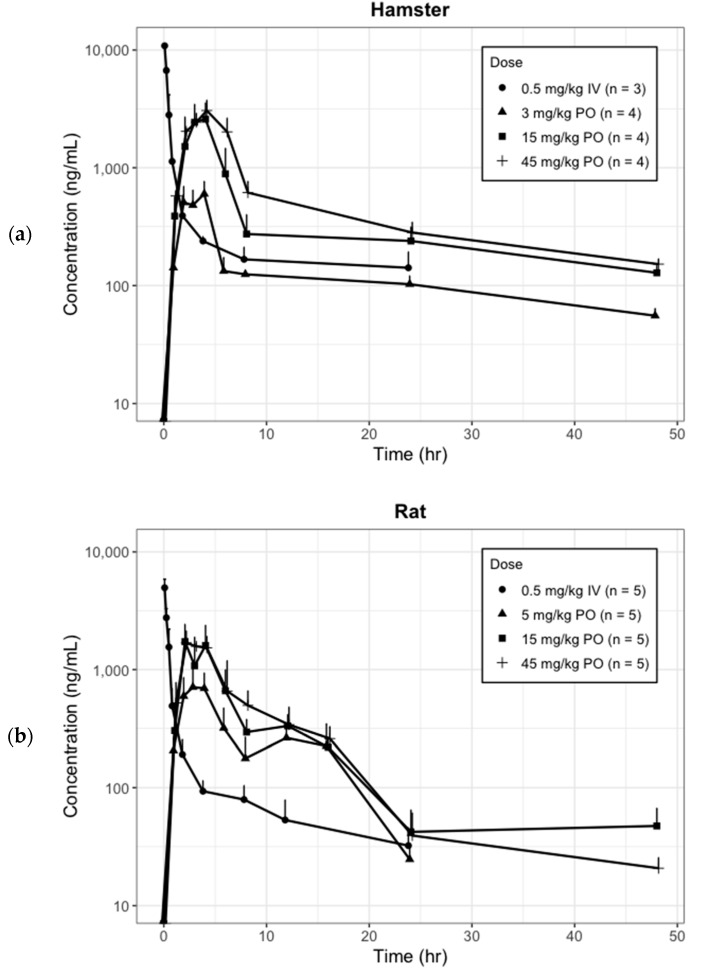
Observed mean plasma concentration (log scale)-time profile of CKD519 by species and dosage regimens (error bar represents the standard deviation). (**a**) Hamster; (**b**) Rat; (**c**) Monkey.

**Figure 2 pharmaceutics-11-00336-f002:**
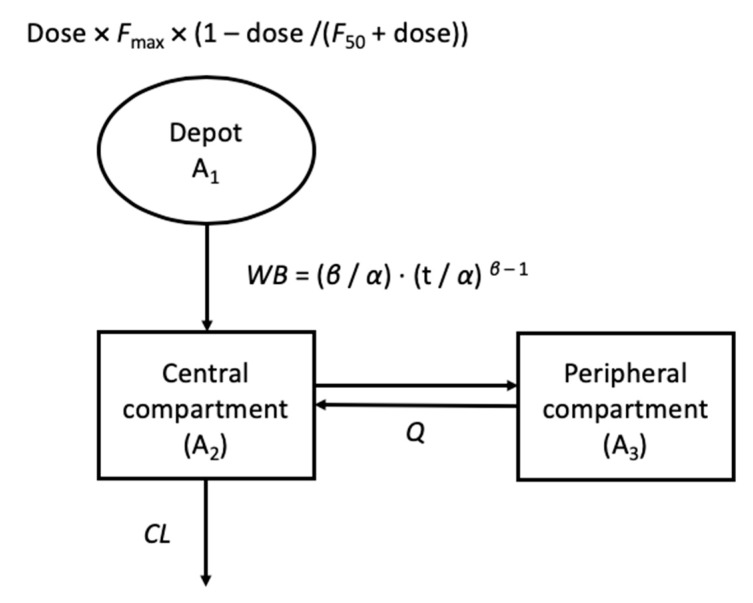
Final pharmacokinetic(PK) structure developed based on animal PK, where *F*_max_ and *F*_50_ is the maximum bioavailability and the amount of dose reaching 50% of the maximum bioavailability, respectively; WB is the time-dependent absorption rate constant following the Weibull distribution *α* as the scale and β as the shape parameters of the Weibull distribution, respectively; A_1_, A_2,_ and A_3_ are the drug amount in the absorption, central, and peripheral compartment, respectively; and *CL* and *Q* are the elimination and distribution clearance, respectively.

**Figure 3 pharmaceutics-11-00336-f003:**
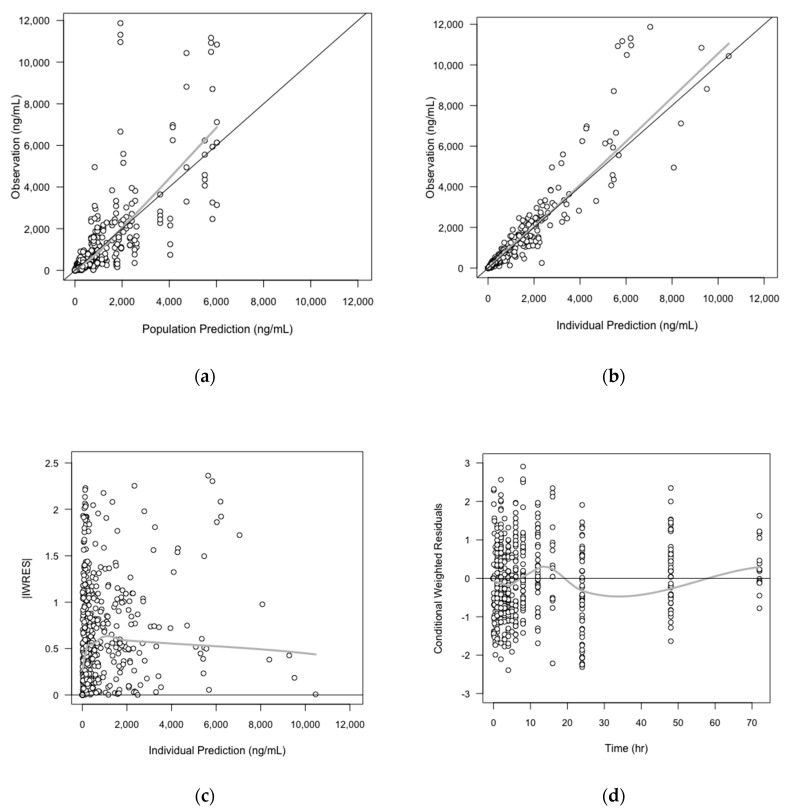
Basic goodness-of-fit (GOF) plots of overall (interspecies) PK model. The solid grey y = x or y = 0 lines are included as a reference. The bold gray lines are the LOWESS (local regression soother) trend lines: (**a**) Population prediction versus observation; (**b**) Individual prediction versus observation; (**c**) Individual prediction versus the absolute value of individual weight residuals (IWRES); (**d**) Time versus conditional weighted residuals (CWRES).

**Figure 4 pharmaceutics-11-00336-f004:**
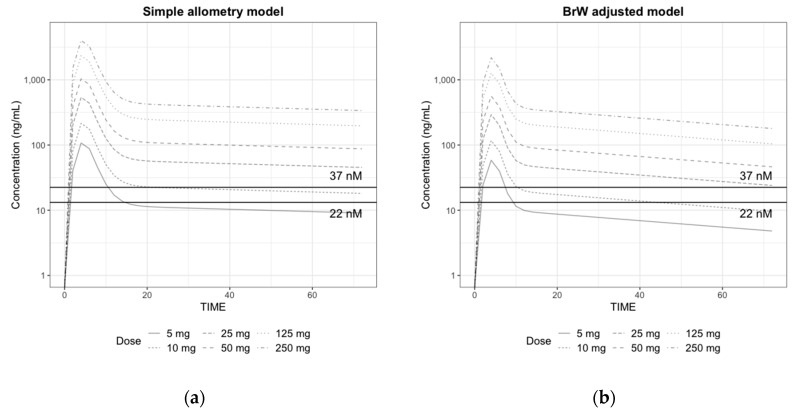
Prediction of plasma concentration in log scale after a single oral dose of 5, 10, 25, 50, 125, and 250 mg of CKD519 in a 60 kg human using the (**a**) predicted PK parameters from simple allometry model and the (**b**) predicted PK parameters from the BrW-adjusted model (horizontal lines represent the IC_50_ (22 nM and 37 nM), respectively)

**Figure 5 pharmaceutics-11-00336-f005:**
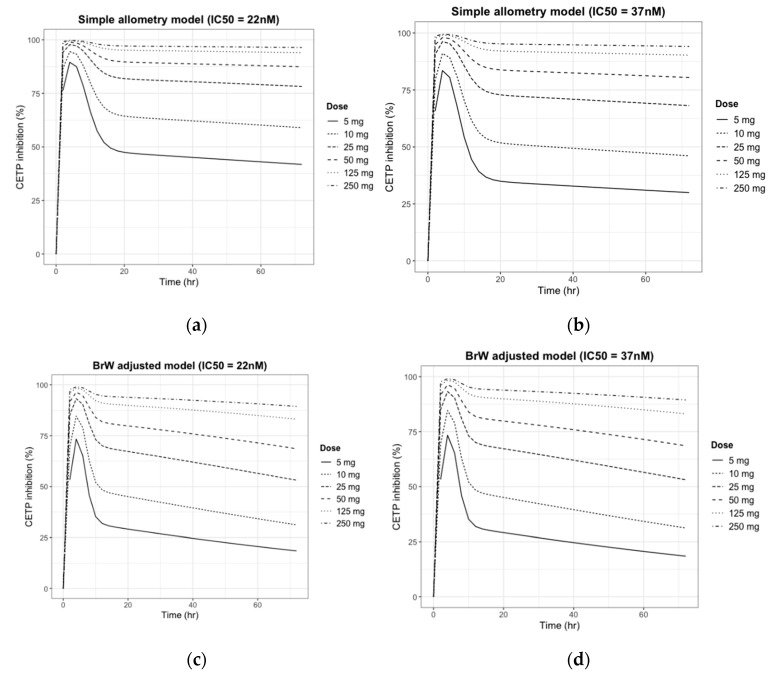
Prediction of CETP inhibition (%) change after a single oral dose of 5, 10, 25, 50, 125, and 250 mg of CKD519 in a 60 kg human using four sets of PK/PD parameters: (**a**) Simple allometry model and reported PD parameter of comparator drug; (**b**) Simple allometry model and predicted PD parameter using in vitro potency ratio; (**c**) BrW-adjusted allometry model and reported PD parameter of comparator drug; and (**d**) BrW-adjusted allometry model and predicted PD parameter using the in vitro potency ratio.

**Figure 6 pharmaceutics-11-00336-f006:**
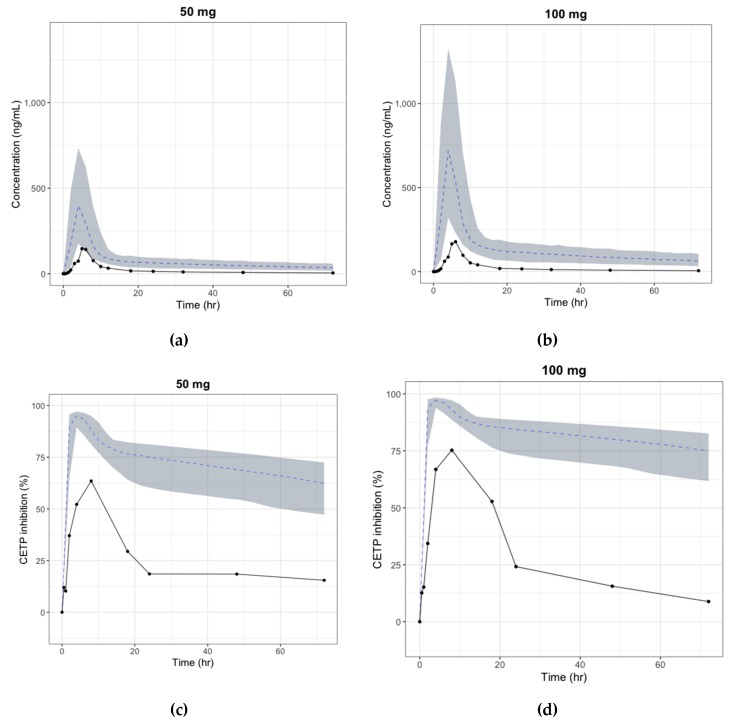
Predicted vs observed time-plasma concentration and CETP inhibition (%) profile in human, respectively (black line: median curve for observation, gray area: 90% confidence interval for simulation, blue dashed: median curve for simulation) (**a**) Plasma concentration change after single oral dose of 50 mg; (**b**) Plasma concentration change after single oral dose of 100 mg; (**c**) CETP inhibition (%) change after single oral dose of 50 mg; (**d**) CETP inhibition (%) change after single oral dose of 100 mg.

**Figure 7 pharmaceutics-11-00336-f007:**
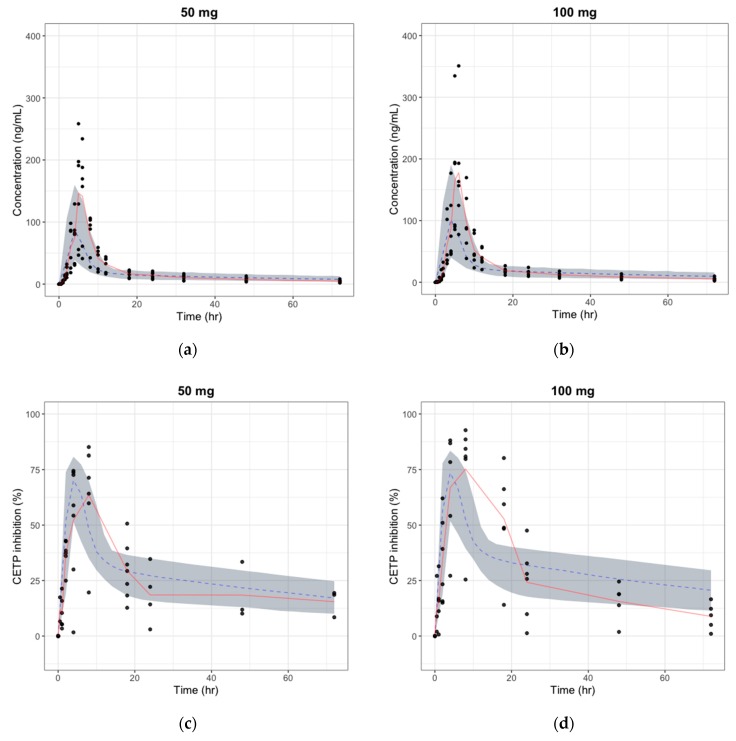
The predicted vs. observed time–plasma concentration and CETP inhibition (%) profile in human after modifying the absolute bioavailability, respectively (red line: median curve for observation, gray area: 90% confidence interval for simulation, blue dashed: median curve for simulation). (**a**) Plasma concentration change after a single oral dose of 50 mg; (**b**) Plasma concentration change after a single oral dose of 100 mg; (**c**) CETP inhibition (%) change after a single oral dose of 50 mg; and (**d**) CETP inhibition (%) change after a single oral dose of 100 mg.

**Table 1 pharmaceutics-11-00336-t001:** Types and values of physiological factors by species.

Species	Weight (kg)	Brain Weight (g)	Maximum Life Span (years)
Hamster	0.15	1.4	2.7
Rat	0.25	1.8	4.7
Cynomolgus monkey	3	74	22.3
Human	60	1500	93.4

**Table 2 pharmaceutics-11-00336-t002:** The in vivo pharmacokinetic properties of CKD519 summarized as the results from non-compartmental analysis.

Species	Administration Route	Sample Size	Sex	Dose (mg/kg)	*C_max_* (ng/mL)	*T*_max_ (h)	*AUC*_last_ (ng∙h/mL)	*CL/F* * (mL/h)	*V*/F * (mL)	*t*_1/2_ (h)	*k*_e_ (h^−1^)
Hamster	Intravenous	3	F	0.5		0.083 (0.083–0.083)	8760 ± 635	6.36 ± 1.20	138 ± 89.9	16.9 ± 14.1	0.0616 ± 0.0446
	Oral	4	F	3	659 ± 89.3	3.5 (3–4)	6140 ± 471	34.7 ± 3.08	1570 ± 163	31.5 ± 4.41	0.0223 ± 0.00294
		4	F	15	3250 ± 465	4 (3–4)	18800 ± 4310	66.6 ± 20.1	2460 ± 162	26.8 ± 5.23	0.0268 ± 0.00613
		4	F	45	3160 ± 614	4 (3–6)	26700 ± 2330	145 ± 9.52	4330 ± 572	20.7 ± 1.76	0.0338 ± 0.00311
Rat	Intravenous	5	F	0.5		0.083 (0.083–0.083)	3940 ± 585	28.9 ± 6.24	494 ± 126	12.0 ± 2.98	0.0677 ± 0.0214
	Oral	5	F	5	893 ± 231	3 (3–4)	6640 ± 932	188 ± 28.6	2860 ± 1930	10.0 ± 4.78	0.0784 ± 0.0245
		5	F	15	2040 ± 448	2 (2–4)	11600 ± 3240	323 ± 82.8	4770 ± 1740	10.1 ± 1.82	0.0701 ± 0.0107
		5	F	45	184 ± 235	3 (2–5)	12700 ± 1900	887 ± 140	10100 ± 1380	7.97 ± 0.782	0.0876 ± 0.00791
Monkey	Intravenous	4	F	0.1		0.1665 (0.083–0.25)	5150 ± 1530	33.5 ± 20.3	3500 ± 1380	104 ± 80.0	0.0105 ± 0.00836
	Oral	4	F	1	752 ± 148	5.5 (5–6)	8940 ± 954	230 ± 64.6	17600 ± 4000	59.9 ± 35.5	0.0140 ± 0.00570
		4	F	5	4700 ± 1100	7 (6–8)	29500 ± 4840	309 ± 77.3	50500 ± 16900	125 ± 64.6	0.00707 ± 0.00412
		4	F	30	10200 ± 2390	8 (8–8)	79100 ± 15600	999 ± 123	67800 ± 44900	46.3 ± 26.8	0.0194 ± 0.0104

All values are presented as mean ± standard deviation except those for *T*_max_, which are the median (range); * For intravenous administration, CL and V were the estimated *C*_max_, maximum plasma concentration; Tmax, time to reach maximum plasma concentration; *AUC*_last_, area under the plasma concentration–time curve from time zero to time of the last measurable concentration; *CL/F*, oral clearance; *V/F*, apparent volume of distribution after oral administration; *t*_1/2_, half-life; *k*_e_, elimination rate constant.

**Table 3 pharmaceutics-11-00336-t003:** Final pharmacokinetic parameter estimates (%RSE) for the hamster, rat, and monkey.

Species	*CL* (L/h)	*V*_c_ (L)	*Q* (L/h)	*V*_p_ (L)	*α*	*β*	*F* _max_	*F* _50_
Hamster	0.00377 (7.40%)	0.00472 (9.43%)	0.00617 (10.3%)	0.0924 (9.78%)	4.20 (4.14%)	2.62 (4.47%)	0.129 (11.3%)	12.8 (16.7%)
Rat	0.0326 (5.58%)	0.0287 (6.41%)	0.0404 (12.0%)	0.285 (8.21%)	3.86 (3.94%)	2.59 (4.59%)	0.43 (18.5%)	5.09 (23.2%)
Monkey	0.0344 (24.0%)	0.383 (74.7%)	0.0895 (49.9%)	3.40 (51.5%)	5.44 (11.8%)	3.54 (6.50%)	0.18 (27.4%)	12.3 (35.4%)

*CL*, clearance; *V_c_*, distribution volume in the central compartment; *Q*, intercompartment clearance; *V_p_*, distribution volume in the peripheral compartment; *F_max_*, maximum bioavailability; *F_50_*, amount of dose reaching 50% of the maximum bioavailability; %RSE value was estimated by the $COV method of NONMEM.

**Table 4 pharmaceutics-11-00336-t004:** List of interspecies structures and corresponding objective function values.

Structures	Objective Function Value(OFV)
*CL(Q)* = a·BW^b^	4670.56
*CL(Q)* = a·BW^b^·BrWc	5226.38
*CL(Q)* = a·BrW^b^/MLP	5263.84
*CL(Q)* = a·BrW^b^/BW	5419.25
*CL(Q)* = a·BW^b^/BrW	4608.67
*CL(Q)* = a·BW^b^/MLP	4902.96

*CL(Q)*, elimination and distribution clearance; BW, body weight; BrW, brain weight, MLP, maximum lifespan potential; a, allometric coefficient; b, allometric exponent for the first variable; c, allometric exponent for the second variable.

**Table 5 pharmaceutics-11-00336-t005:** Final parameter estimates and bootstrap outcome for the CKD-519 overall (interspecies) PK model.

Parameter	Description	Simple Allometry		Brain Weight-Corrected	
		Estimates	Bootstrap Median (90% Confidence Interval(CI))	Estimates	Bootstrap Median (90% Confidence Interval(CI))
*CL* * = θ_1_ × WT ^θ2^ (/BrW) *					
θ_1_	Coefficient for *CL*	0.0308	0.0308 (0.0259–0.0379)	0.488	0.490 (0.420–0.588)
θ_2_	Exponent for *CL*	0.525	0.526 (0.469–0.586)	1.90	1.90 (1.83–1.95)
*V*_c_ = θ_3_ × WT^θ4^					
θ_3_	Coefficient for *V*_c_	0.0755	0.0763 (0.0628–0.0943)	0.0735	0.0756 (0.0626–0.0954)
θ_4_	Exponent of *V*_c_	1 (fix)		1 (fix)	
*Q* = θ_5_ × WT^θ6^ (/BrW) *					
θ_5_	Coefficient for *Q*	0.0549	0.0553 (0.0392–0.0760)	0.846	0.837 (0.595–1.16)
θ_6_	Exponent of *Q*	0.670	0.663 (0.504–0.833)	2.04	2.03 (1.85–2.21)
*V*_p_ = θ_7_ × WT^θ8^					
θ_7_	Coefficient for *V*_p_	0.829	0.832 (0.724–0.999)	0.813	0.813 (0.719–0.960)
θ_8_	Exponent of *V*_p_	1 (fix)		1 (fix)	
*WB* = (*β*/*α*) × (t/*α*) *^β^* ^− 1^					
α	Scale parameter of *WB*	4.50	4.50 (4.21–4.80)	4.51	4.49 (4.21–4.80)
β	Shape parameter of *WB*	3.01	3.10 (2.89–3.32)	3.11	3.10 (2.79–3.38)
*F* = *F*_max_ × (1 − dose/(*F*_50_ + dose))					
*F* _max_	Maximum bioavailability	0.183	0.182 (0.151–0.219)	0.192	0.192 (0.159–0.229)
*F* _50_	Amount of dose reaching 50% of the maximum bioavailability	11.9	12.4 (9.55–16.2)	10.6	10.8 (8.89–14.8)
*ω*_CL_ (%)	Interindividual variability of *CL*	42.8	42.2 (35.9–47.4)	25.9	25.2 (19.8–31.4)
*ω*_Q_ (%)	Interindividual variability of *Q*	23.9	23.5 (19.5–26.6)	60.8	0.579 (0.357–0.727)
*ω*_α_ (%)	Interindividual variability of *α*	27.1	26.5 (21.3–31.8)	27.0	23.6 (8.40–27.6)
*ω*_β_ (%)	Interindividual variability of *β*	65.7	62.4 (47.3–76.3)	24.0	26.2 (3.00–34.1)

BW, body weight; BrW, brain weight; MLP, maximum lifespan potential; *CL*, clearance (L/h); *Vc*, distribution volume in the central compartment (L); *Q*, distribution clearance (L/h); *Vp*, distribution volume in the peripheral compartment (L); *WB*, the time dependent absorption rate constant following Weibull distribution(1/h); *F*, absolute bioavailability; * reflected according to the allometry structure.

**Table 6 pharmaceutics-11-00336-t006:** Predicted human pharmacokinetic/pharmacodynamic(PK/PD) parameters of CKD-519 (60 kg).

Allometry	*CL* (L/h)	*V*_c_ (L)	*Q* (L/h)	*V*_p_ (L)	*α*	*β*	*F* _max_	*F* _50_
Simple	0.264	3.29	1.17	49.7	4.51	3.11	0.192	10.6
Brain weight-corrected	0.777	4.41	2.39	48.8				

*CL*, clearance; *V_c_*, distribution volume in the central compartment; *Q*, distribution clearance; *V_p_*, distribution volume in the peripheral compartment; α as the scale and β as the shape parameters of the Weibull distribution, respectively; *F_max_*, maximum bioavailability; *F_50_*, amount of dose reaching 50% of the maximum bioavailability.

**Table 7 pharmaceutics-11-00336-t007:** Absolute bioavailability estimated by calculating the AUC ratio between the observed oral PK profile and the predicted intravenous PK profile in humans.

Dose (mg)	Absolute Bioavailability
25	0.0362
50	0.0321
100	0.0191
200	0.0140
400	0.00814

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
