# Peer review of "Quantitative Prediction of Human Pharmacokinetics and Pharmacodynamics of CKD519, a Potent Inhibitor of Cholesteryl Ester Transfer Protein (CETP)"

_pharmaceutics, 2019, doi:10.3390/pharmaceutics11070336_

Round 1

Reviewer 1 Report

Many comments in the revision have been satisfactorily addressed. Please find some comments below:

- Specify what to the error bars in Fig.1 represent

- Please provide precision of parameter estimates (i.e. CV%) in Table 3 

- From figure 7, it seems that the maximum CETP% inhibition is predicted to be earlier than what is observed. Did the authors consider any delayed effects  model (biophase, indirect response model)? This is also seen for the plasma concentrations , hence alternatively were any delay in absorption considered?

Reviewer 2 Report

Comments to the authors:

Overall this manuscript has been drafted well. However, the resolutions of figures might need to be increased as they look very blurry. My other suggestion would be indicate the threshold concentrations in the simulations i.e how long does the concentration remain over the IC50? It would be helpful to understand and interpret the proposed dosing regimen.

Table 2: Please put * for to show difference for CL/F and V/F in IV vs Oral administration and NCA results.

Figure 5: Please show the threshold concentration i.e IC50 or IC90

Why are the authors assuming the clearance to be linear? Can there some enzyme induction which can lead to faster clearance at high doses leading to decreased F? Have the authors used other non-linear models to fit this data? 

Specific Comments/Questions:

Materials and Methods:

It would be helpful to understand and interpret these results if the authors can comment on the formulations which were used for the different species and the final human study. Where there any formulation differences between various species? Also, it would be helpful to specify if the drug was administered in fasted or fed state.

  Overall strategy (line 85):

Authors state that Human PK/PD modeling and simulation incorporating PD information of the comparator:

Question: Did the authors use variability (%cv) for PK and PD parameters used in the simulations? Please specify the assumptions for POC simulations in the manuscript.

Line 145-46:

“To address the uncertainty, two allometric relationships were selected as final models, and two sets of human PK parameters were predicted—one from the allometry of best fit and the other from the simple allometry which is most conventional.”

Suggestion: Please explain clearly the two approaches. “Simple allometry” can be interpreted in many ways.

Line 148-149:

“Human values for absorption parameter were determined not by the allometric method but by the final estimates from the overall PK model.”

Question: Why was allometric scaling not used to determine this rate of absorption? Can the authors elaborate on this so that it is clear to the readers?

Line 160

“The peak effect was around ~90% inhibition. Since no in vivo information for the IC50 value of CKD-519 was available, it was assumed using the known in vivo IC50 value for the comparator (22 nM) [17], and the observed difference in the in vitro potency between the comparator and CKD519 (IC50 1.3 nM vs 2.3 nM (in-house data), respectively).”

Question:

What about the potency of other drugs in this class? What is the rationale behind using Merck’s drug? If the authors can also look at the in-vitro potency and come up with an understanding of the variability, then this can be incorporated into the simulations which in my opinion would be ideal for POC kind of simulations and translational modeling.

Authors state that “ The human PK parameter values  for the comparator obtained from the published literature showed a similar trend [13-17]”

Suggestions: It would be valuable for the readers if authors can provide the ranges for these PK parameters or whichever is the closest to this molecule.

  Line 360:

Authors state: “As same-in-class drugs generally follow a similar pharmacodynamic pathway with little difference in potency [22]”

Question: What do the authors mean by little difference? Assuming certain variability around the potency will be help to come with the dose for POC simulation.

Author Response

This manuscript is a resubmission of an earlier submission. The following is a list of the peer review reports and author responses from that submission.

Round 1

Reviewer 1 Report

The manuscript describes allometry-based approach to predict the human pharmacokinetics and pharmacodynamics of CKD519, a selective inhibitor of CETP, based on animal experimental data. The researchers used data from 3 species (hamsters, rats and monkeys), different pharmacokinetic models, and different allometric approaches to determine the drug dose to be used in Phase I clinical studies. This research project has a technical nature, which limits its novelty. In addition, it has several flaws in the presentation, discussion and interpretation of the experimental results, as outlined below:

Major comments:

1. The modeling-based predictions were compared to the Phase I data (see Fig. 8). The abstract fails to mention that, and that the applied models substantially (several-fold) over-estimated the CDK519 concentrations and effects. Description of this Phase I clinical study and its data should appear in the Methods and Results (and not just in the Discussion).

2. What was the source of the pre-clinical PK and PD data? How many animals were used? What was their gender?

What was the origin of the Phase I clinical data (Fig. 8)? How many subjects participated in this study?

What is the relationship of the authors to the Chong Kun Dang Pharmaceutical Company that investigates CDK519? Why it has not been disclosed in the manuscript?

3. The presentation of the pre-clinical data is partial only (Fig. 2); Table 1 lists only part of the PK parameters that are not representative. Please list the major PK parameters in the individual studies, including the F, V, CL, t1/2, AUC values, etc. Please report these values as averages and standard deviations; please add the error bars to Fig. 2.

4. The description of the allometric approaches and their results is very partial, and it is difficult to understand the performance of the individual approaches, and the choices of the “best” models by the researchers. E.g. Table 2 lists only the OFV values, without the fits or deviations of the models. Some of the assumptions and results are questionable, e.g.:

 a. assumption of similar absorption in humans & animals (line 141) – please show the parameters relevant to the drug absorption (F, ka,  first-pass effect) in each one of the species.

 b. The allometric exponent b value (Table 5) are very far from 0.75 – the “classical” value of this exponent in many studies.

 c. is the CEPT expression levels & its contribution to the metabolism different in the studied species and the humans? Please present these data and relevant citations.

Minor issues:

Page 2, line 58: Anacetrapib clinical development has been abandoned in 2017.

Fig. 8 c-d: please correct the y axes labels - % inhibition.

Author Response

Response to Reviewer 1 Comments

1.     The modeling-based predictions were compared to the Phase I data (see Fig. 8). The abstract fails to mention that, and that the applied models substantially (several-fold) over-estimated the CDK519 concentrations and effects. Description of this Phase I clinical study and its data should appear in the Methods and Results (and not just in the Discussion).

Response 1: Since phase 1 study was not conducted in our institution and the result has not been published yet, by sponsor’s principle, the details of the data are not allowed to be disclosed unfortunately.  Currently, it is our best to suggest the result as a graph to a discussion section only for comparing the general trend of data. For this reason, I could not reflect your requirement on the manuscript even though it would be very informative. I feel genuinely sorry about this issue and I sincerely hope for the favor of your understanding

2.     What was the source of the pre-clinical PK and PD data? How many animals were used? What was their gender?

Response 2: Pre-clinical PK and PD data was provided by ChongKunDang, Korea. The number and sex of animals used for analysis was added in Table 1 as requested. 

2.1.What was the origin of the Phase I clinical data (Fig. 8)? How many subjects participated in this study?

Response 2-1: The PK profile of Phase I clinical data was also provided by ChongKunDang, Korea. It was a single ascending study in healthy Koreans which was conducted by another institution and the ownership of data is on ChongKunDang. Therefore, it’s not possible to disclose any other results or information than current graph. I totally understand that information that reviewer mentioned would be more useful than showing the graph. However as mentioned above, it was the sponsor's principle, and I feel sincerely sorry about this issue.  

2.2.What is the relationship of the authors to the Chong Kun Dang Pharmaceutical Company that investigates CDK519? Why it has not been disclosed in the manuscript?

Response 2-2: This analysis was conducted by authors affiliated to Qfitter and PIPET (Pharmacometrics Institute for Practical Education and Training). They were supported by ChongKunDang for the pharmacometrics analysis of CKD519. This information was added to the Acknowledgement section for clarification. 

3.     The presentation of the pre-clinical data is partial only (Fig. 2); Table 1 lists only part of the PK parameters that are not representative. Please list the major PK parameters in the individual studies, including the F, V, CL, t1/2, AUC values, etc. Please report these values as averages and standard deviations; please add the error bars to Fig. 2.

Response 3: As the objective of this study was to estimate the parameters of compartmental PK models by mixed effect modeling method for predicting human parameters, NCA parameters were used for exploratory purpose only. Therefore, in this study, it is considered that finding and listing all of the NCA parameters is not meaningful. For the parameters listed in Table 1, their average and standard deviation were specified as requested. Also, we added error bars to Figure 2 (Currently, Figure 1)as requested. 

4.     The description of the allometric approaches and their results is very partial, and it is difficult to understand the performance of the individual approaches, and the choices of the “best” models by the researchers. E.g. Table 2 lists only the OFV values, without the fits or deviations of the models. Some of the assumptions and results are questionable, e.g.:

Response 4: After the reviewers’ comment, we revised the overall manuscript to modele all animal PK data simultaneously and developed the interspecies model instead of modeling each animal data by species and estimating parameters by species. For exploratory analysis, we assessed the possibility of developing integrated interspecies model by modeling each animal data by species and based on that all three animal data were well described by a single model, we developed the integrated interspecies model. The final interspecies model was selected based on the OFV value and visual inspection such as goodness-of-fit plots (Presented in Table 3 and Figure 3), and the precision of parameters were assessed by confidence intervals from the bootstrap analysis (Presented in Table 4). 

4.1.assumption of similar absorption in humans & animals (line 141) – please show the parameters relevant to the drug absorption (F, ka, first-pass effect) in each one of the species. 

Response 4-1: In the absence of clinical data on humans, as all 3 animal species were described well by an integrated model, we assumed that the absorption pattern in human is similar to the absorption pattern in the integrated animal model. Therefore, estimated absorption parameter values of interspecies model were used as those of human model. Since we modeled the i.v. data and oral data simultaneously, the F value presented in Table 4 and 7 is an absolute bioavailability, which is also referred as first-pass effect. In this study, the F was estimated to be a value that varies with oral doses in all animal species when the F in i.v. dose is fixed to 1. Thus, as mentioned above, we developed the interspecies model in order to find the integrated absorption pattern and reflect it in the human species, all animal species were simultaneously analyzed to predict integrated F equation, which is specified as the Equation (4). In this study, Weibull type absorption model was used to describe the data, and in weibull type absorption model, absorption rate was described by alpha and beta, not Ka, which are listed in Table 4 and 7.   

4.2.The allometric exponent b value (Table 5) are very far from 0.75 – the “classical” value of this exponent in many studies.

Response 4-2: The classical allometric exponent corresponds to 0.75 when simple the allometry method is used. As we used both methods (Simple allometry method and brain weight adjusted method), exponents derived from the simple allometry method are listed in Table 6, which are acceptable range (0.425, 0.567) based on literatures. However, from ‘Rule of exponent’, when exponent of simple allometry scaling is less than 0.55, parameters can be underestimated when simple allometry scaling is used. Thus, correction by brain weight or maximum life span can be considered, and the coefficient of determination(R2) of parameters from three species can be used to assess the correlation of parameters. As mentioned in method 3.1 (From Line 200 to 221), incorporating the product of BrW and parameter into the simple allometric scaling model showed strongest correlations (R2= 0.9760, R2= 0.9993, respectively) compared to simple allometric scaling (R2= 0.5229, R2= 0.8896, respectively). In this case, the range for the allometric exponent can be wider than that from conventional allometry scaling method, and also exponent of allometry scaling is considered to be not universe but dependent on data. (Reference: (1) Iftekhar Mahmood (2018) Misconceptions and issues regarding allometric scaling during the drug development process, Expert Opinion on Drug Metabolism & Toxicology, 14:8, 843-854, DOI: 10.1080/17425255.2018.1499725, (2) Peter L. Bonate, Danny R. Pharmacokinetics in drug development : clinical study design and analysis, page 423-432 (3) Huh, Y., Smith, D. E., & Feng, M. R. (2011). Interspecies scaling and prediction of human clearance: comparison of small- and macro-molecule drugs. Xenobiotica; the fate of foreign compounds in biological systems, 41(11), 972-87.) 

4.3.is the CEPT expression levels & its contribution to the metabolism different in the studied species and the humans? Please present these data and relevant citations.

Response 4-3: In this study, as CETP inhibition level data in animal species was not available, we did not use it for predicting the PK/PD profile. Instead, the best possible way to predict human PK/PD profile without in vivo data was to compare the in vitro IC50 values of the comparator drug and CKD 519, which were measured in identical experiment condition, and extrapolate it to human using developed PD model of comparator drug in literature. Thus, consideration of the difference in CETP expression levels and its contribution between species was not done in this study. 

Minor issues:

Page 2, line 58: Anacetrapib clinical development has been abandoned in 2017.

We have modified the main text (line 57) as requested. (Marked as highlighted) 

Fig. 8 c-d: please correct the y axes labels - % inhibition.

We have modified the Figure 8 as requested

Reviewer 2 Report

In this paper, the authors have integrated in vitro and in vivo data along with available competitor data for PK/PD analysis and model development for CKD519, a selective inhibitor of CETP. While the approach of model-informed drug development is appreciated, the manuscript needs several major revisions to be considered for publication. There are shortcomings in the experimental studies, presentation of the results, model evaluation and final model simulations and comparison to clinical data. Please find below specific comments for the consideration of authors:

-          Several times throughout the manuscript the authors interchangeably use ‘PK/PD profile’ and ‘PK/PD model’ . The distinction is important. For example, in the abstract, line 7 the authors state “…to predict human PK model”. The prediction is for the human PK profile. The model is not predicted. Another example is in line 75 where in the authors have said “animal PK model and parameters were developed and predicted”

-          The references cited for the utility in PK/PD modeling need to be updated. For example, reference no.28 is from 2008. More recent references should also be included.

-          The overall strategy is not very clear in the manuscript – why are PK parameters predicted twice? What is the rationale? Where model derived PK parameters used or not? Was single species scaling or multiple species scaling was used? If multiple species scaling was used, please present the relationship of parameters across all three species in a graphical and mathematical equation format. The diagram is also not designed well (Figure1) -Why is there no link between the PK model and parameters?

-          The number of animals per dose group should be mentioned in section 2.2

-          Please list standard deviation in Table 1 for PK Parameters

-          Please list the precision on the PK model parameter estimates in Table 3

-          Please clarify what is presented in the graphs in Figure 2. Please consider presenting it as mean with standard deviation as error bars

-          Figure 4 is for which species? Please specify. Consider showing results for all species.

-          Please show at least representative model fits to the PK profile with individual and population model fits overlaid

-          Though the model selection and evaluation criteria is defined well, the goodness of fit plots do show some bias. Please present the CWRES plots again population prediction for better evaluation.

-          Please consider modeling all data simultaneously.

-          It is not clear if the IV data was used for model development; if not the I recommend that the authors use that data to model the IV and oral data simultaneously. This will also enable to estimate the true CL and to determine the non-linear changes in CL vs. F

-          Figure 8 highlights stark deviation between observed and model-predicted data. The authors could discuss this further in the discussion section. Additionally, it would also be useful to see the comparison between predicted and observed human PK parameters and devise better prediction strategies.

Author Response

Response to Reviewer 2 Comments

Inthis paper, the authors have integrated in vitro and in vivo data along with available competitor data for PK/PD analysis and model development for CKD519, a selective inhibitor of CETP. While the approach of model-informed drug development is appreciated, the manuscript needs several major revisions to be considered for publication. There are shortcomings in the experimental studies, presentation of the results, model evaluation and final model simulations and comparison to clinical data. Please find below specific comments for the consideration of authors:

1.    Several times throughout the manuscript the authors interchangeably use ‘PK/PD profile’ and ‘PK/PD model’ . The distinction is important. For example, in the abstract, line 7 the authors state “…to predict human PK model”. The prediction is for the human PK profile. The model is not predicted. Another example is in line 75 where in the authors have said “animal PK model and parameters were developed and predicted”

Response 1: I sincerely agree with your point. For distinction between ‘PK/PD model’ and ‘PK/PD profile’, we have modified the abstract and main text as requested and highlighted the modified part. 

2.    The references cited for the utility in PK/PD modeling need to be updated. For example, reference no.28 is from 2008. More recent references should also be included.

Response 2: The additional references were added as requested. (Reference 25, 27, 28, 32, and 33)

3.    The overall strategy is not very clear in the manuscript – why are PK parameters predicted twice? What is the rationale? Where model derived PK parameters used or not? Was single species scaling or multiple species scaling was used? If multiple species scaling was used, please present the relationship of parameters across all three species in a graphical and mathematical equation format. The diagram is also not designed well (Figure1) -Why is there no link between the PK model and parameters?

Response 3: The PK parameters were estimated using all three animal species parameters, and because it is the empirical model, two methods were used to reduce uncertainty. First was most conventional method (Simple allometry) and the second was the fittest method in all three animal species based on OFV value (Brain weight adjusted allometry). For small molecules, there is uncertainty of using interspecies scaling method. Thus, we tried to find the dose which satisfies the condition in both methods. Because there were only three animal species, showing regression plot doesn’t seem meaningful when there’s coefficient of determination value in the manuscript. Thus, instead of showing in graph, we decided to add the table 5 to show the both PK parameters estimated by integrated interspecies model and single species model and the relationships of parameters used for interspecies model are presented in Table 4. The mathemathical equation format is presented in Table 4. We decided to remove the Figure 1. Instead, we trimmed the structure of method section to enhace the reader’s understanding.

4.    The number of animals per dose group should be mentioned in section 2.2

Response 4: The number of animals used for analysis was added in Table 1 as requested.

5.    Please list standard deviation in Table 1 for PK Parameters

Response 5: Standard deviation was added in Table 1 as requested. 

6.    Please list the precision on the PK model parameter estimates in Table 3

Reponse 6: Bootstrap method was performed for precision assessment and the confidence intervals were added in Table 4. 

7.    Please clarify what is presented in the graphs in Figure 2. Please consider presenting it as mean with standard deviation as error bars

Response 7: Legend of Figure 2 (Currently figure 1) was modified for clarification, and error bars were added for presenting the standard deviation.

8.    Figure 4 is for which species? Please specify. Consider showing results for all species.

Response 8: Based on reviewer’s opinion, we decided to model simultaneously. Thus, we presented the representative model fits using interspecies model which was modeled all data simultaneously. 

9.    Please show at least representative model fits to the PK profile with individual and population model fits overlaid

Response 9: We presented the representative model fits of interspecies model in Figure 3. 

10.  Though the model selection and evaluation criteria is defined well, the goodness of fit plots do show some bias. Please present the CWRES plots again population prediction for better evaluation.

Response 10: As intersubject variability was not applied due to the relatively small number of animals, individual prediction showed some bias in the goodness of fit plots. However, as the precision was reasonable based on bootstrap results, we considered that estimated parameters were reliable. Because individual prediction and population prediction are identical in this model, we removed the redundant figure. Instead, we added the CWRES plots against population prediction in Figure 3 as requested. 

11.  Please consider modeling all data simultaneously.

Response 11: Considering reviewer’s opinion, we decided to model all data simultaneously for the integrated interspecies model and parameters were estimated from the model. Thus, we modified the method and result section accordingly. (Marked as highlighted) 

12.  It is not clear if the IV data was used for model development; if not the I recommend that the authors use that data to model the IV and oral data simultaneously. This will also enable to estimate the true CL and to determine the non-linear changes in CL vs. F

Response 12: In this study, i.v. and oral data were used simultaneously, and true CL and F (Absolute bioavailability) were predicted. From exploratory analysis, there were non-linear changes in F by doses, so the parameters (F50, Fmax) were estimated to express the F value, and it is represented as Equation (8).

13.  Figure 8 highlights stark deviation between observed and model-predicted data. The authors could discuss this further in the discussion section. Additionally, it would also be useful to see the comparison between predicted and observed human PK parameters and devise better prediction strategies.

Response 1: Since phase 1 study was not conducted in our institution and the result has not been published yet, by sponsor’s principle, the details of the data are not allowed to be disclosed. Currently, it is our best to suggest the result as a graph to a discussion section only for comparing the general trend of data. For this reason, I could not reflect your requirement on the manuscript even though I totally understand that it would be more useful than showing the graph. I feel genuinely sorry about this issue and I sincerely hope for the favor of your understanding. 

I agree that I should discuss this issue further for better prediction strategies. I added more content about the devidation between observed and model-predicted data to the discussion part as you mentioned and marked as highlighted. 

Reviewer 3 Report

I have attached my comments. Please find the attached file.

Round 2

Reviewer 1 Report

The authors made substantial corrections and additions to the manuscript. The connection of the authors and of the study to the Chong Kun Dang Company has been also clarified.

Still, the title, abstract and other parts of the manuscript fail to mention that the prediction of human data based on pre-clinical data was largely unsuccessful.

The fits of the developed “interspecies” model (observed vs. predicted plots) are not shown. It is hard to understand whether this model described appropriately the observed PK data.

Additional comments:

The overall number of animals used in the study  - does not match for the hamsters (compare to Table 1).

Table 1 - please correct the number of significant digits.

Some proofing of the text, especially of the newly added parts, is needed to correct terminological and grammatical errors.

Reviewer 2 Report

Thank you for consideration of comments and for incorporating some of them. Here are few comments on the revised version:

- The distinction of PK/PD profile and model is still not made and they are still used interchangeably in the wrong context throughout out the paper. For example in the abstract (line 19) a model is developed, not profile.

- In Table 1, how is standard deviation report for n<3?

- Why was bootstrap analysis done? Was the model not stable enough to provide precision on parameter estimates directly? This is not discussed?

- Figure 1 should be in log-scale 

- The authors indicate that they have provided representative model fits in Figure 3, but they have not. Also, the GOF plots should be vs. population prediction- which is a better indicator of GOF for population models.

- Since the authors are not able to discuss regarding the clinical data, the conclusions regarding predicting the efficacious dose cannot be considered.

- There are several grammatical errors throughout the manuscript, especially after the revision (for example lines 85-91) leading to sentences that are not clear enough to comprehend